# Comparative Analysis of Metal Electrodeposition Rates towards Formation of High-Entropy WFeCoNiCu Alloy

**DOI:** 10.3390/ma17071513

**Published:** 2024-03-27

**Authors:** Tomasz Ratajczyk, Mikołaj Donten

**Affiliations:** Faculty of Chemistry, University of Warsaw, 02-093 Warsaw, Poland

**Keywords:** tungsten, iron, cobalt, nickel, copper, alloys, high-entropy alloy, electrodeposition, induced codeposition, deposition rates

## Abstract

This study presents a calculation and comparison of Fe, Co, Ni and Cu deposition rates in the tungsten codeposition process based on the electrodeposition of numerous tungsten alloys. Eight different tungsten alloys containing from two to five metals were electrodeposited in constant conditions in order to compare the exact reduction rates. The calculated rates enabled control of the alloy composition precise enough to obtain a high-entropy WFeCoNiCu alloy with a well-balanced composition. The introduction of copper to form the quinternary alloy was found to catalyze the whole process, increasing the deposition rates of all the components of the high-entropy alloy.

## 1. Introduction

The electrodeposition of tungsten alloys is an important research niche, yielding various applicative materials due to their corrosion resistance or catalytic properties. Dating from 1931 [1], research on tungsten electrodeposition has led to the development of several kinds of tungsten alloys with metals such as Fe, Co, Ni [2] and Cu [3], which act as *inducing metals* for the reduction in tungstates [4]. Besides tungsten, such alloys must contain one or more inducing metals. There has been no single original work recently that has compared the electrodeposition efficiencies of the most basic binary alloys of tungsten, let alone of materials consisting of three or more components. Most original papers cover only selected binary or ternary alloys, focusing mainly on their applicational properties [5,6,7,8,9,10]. The range of possible applications of tungsten alloys obtained via induced codeposition is broad indeed. The most common applications of electrodeposited tungsten alloys include protective and decorative layers, especially as anti-corrosive protective coatings for steel surface enrichment, a substitute for hard chrome coatings [5,6,7]. Another important application for electrodeposited tungsten alloys is obtaining catalytic layers, mainly for hydrogen evolution [8,9,10]. All the alloys studied in the mentioned works consist of tungsten and one or two other inducing metals. A trend to approach the electrodeposition of materials containing more metallic components is visible in the most recent publications [11,12,13].

An attempt at obtaining an alloy of higher complexity in terms of qualitative composition and controlling it quantitatively should rely on a comparison of deposition rates for the individual metals in the system, mostly the inducing metals. However, the mentioned background makes the comparison difficult and uncertain. The minimum variables necessary to calculate the rates are current density *j*, current (faradaic) efficiency *CE* and exact alloy composition, or at least the atomic content of tungsten *at%* W for binary alloys. In such cases, the deposition rate *v* of a certain element *X* on the normalized area of the cathode, in the *z*-electronic process, can be expressed as *v* = *j* · *CE* · F^−1^ · *z*^−1^ · *at% X* [mol · s^−1^ · m^−2^]. Although *CE* is a quantity that is crucial to describe the deposition process, numerous authors skip it in their publications. Furthermore, most works that contain the needed information present it not as exact numbers, but rather as graphs, which focus on the influence of a particular parameter on the outcome. Many of these parameters, which affect the deposition rates, make it difficult to compare the results of various researchers, especially since each of them conducts the experiments under different conditions, i.e., bath composition (both qualitative and quantitative) and temperature. Thus, despite almost a century of research on tungsten codeposition, the numerical results cannot be easily unified.

In Table 1, several sets of data have been collected concerning the deposition rates *v* of Fe, Co and Ni in binary alloys with W. The numbers were calculated from *j*, *CE* and *at%* W contained in tables or gathered from plots (in the latter case, the source is mentioned with an asterisk). In the absence of more general criteria of choice, the highest achieved rate of Fe/Co/Ni deposition was chosen for each source. All the *v* values [mol · s^−1^ · m^−2^] were multiplied by 10^5^ for legibility.

Although the occurrence of major discrepancies between the rates achieved by separate researchers cannot be denied, the results do overlap partly. Most notably, the *v* of Ni is usually more than an order of magnitude lower compared to Fe and Co. The difference in *j* applied by various authors is the most obvious cause of the inconsistency in the results, but even for the same *j*, the numbers vary due to the divergency of the other plating conditions, such as bath composition and temperature. Nevertheless, the papers cover some important cases of the influence of those parameters on the *v*. Plots of *at%* W and *CE* against *j* are quite common [14,15,16,18,19,20], even though some authors stay at a constant value of *j*, described as optimal [17,21]. Plots of induced metal deposition rates against *j* tend to increase monotonically due to the apparent increase in current passing through the circuit. The increase is concave down though, mostly because of the *CE* decrease and simultaneous increase in the *at%* W in the alloys. Moreover, applying a pulse current was found to possibly increase the *at%* W and *CE*, with a higher *v* of cobalt at higher frequencies of the pulses [17]. Other variables found in the prior papers include plating bath temperature [22] or tungstate concentration [23], which showed no clear correlations with the *v* of nickel.

Introducing ions of an additional inducing metal to the bath, which leads to the deposition of ternary or multinary alloys, has a certain influence on deposition rates for particular metals. The presence of Cu(II) ions in the solution [24,25] inevitably lowers the *at%* of other metals, but increases the *CE* of the whole deposition process, especially in the case of WNiCu alloys. The interpretation of the data contained in previous works is even more difficult when it comes to mixing more than one iron-group metal into the alloy, as most papers on the electrodeposition of ternary tungsten alloys do not cover any comparison to binary alloys. Possible combinations of tungsten electrodepositions along with the inducing metals may eventually lead to the formation of multinary alloys, consisting of four (e.g., WFeCoNi) or five (e.g., WFeCoNiCu) substantial components. Alloys containing at least five metals above 5 at% can be classified as *high-entropy alloys*, according to the original definition coined by Yeh [26]. Materials similar to the quintenary WFeCoNiCu alloy described hereby were published in recent papers [11,12], but they differ substantially in the qualitative composition [11] or alloy morphology and structure [12], a result of utilizing different plating baths.

As mentioned before, more precise control of the deposition rates of particular metals is an essential requirement for obtaining W alloys with more than one inducing metal. To obtain an alloy consisting of three or more metals, the deposition rates for particular metals have to be calculated for the same system at constant plating parameters. The main aim of the following study is to organize the current knowledge on these deposition efficiencies, which may facilitate more precise tuning of the applicational properties of electrodeposited tungsten alloys, and enable further investigation on deposition of multinary alloys of tungsten. Thus, the following work is intended to quantify the rates of Fe, Co, Ni and Cu codeposition with W and to evaluate the ratios between these rates, as a prelude to research on the electrodeposition of multi-component alloys of tungsten, based on the electrodeposition of nanocrystalline WFeCoNiCu high-entropy alloy.

## 2. Materials and Methods

As this study concerns the comparison of deposition rates of the inducing metals in alloys with tungsten, several different materials were obtained at varying concentrations of inducing metal ions, while keeping all the essential conditions constant. All the samples were deposited from a tungstate–citrate aqueous plating bath. All the reagents were analytical grade. Before each experiment, 400 mL portions of the plating baths were freshly prepared by mixing concentrated solutions of the inducing metal salts into a new portion of the tungstate–citrate bath. The solutions were not stirred nor moved in any manner during the electroplating. The composition of the tungstate–citrate bath utilized in the study is enumerated in Table 2. A fresh portion of the plating bath prepared in such a manner has a pH of 8.0–8.2 and was not observed to change substantially during the experiments.

The usage of ferric ammonium citrate deserves some individual attention. For the electrodeposition of iron–tungsten alloys, baths containing iron(II) salts are commonly used, such as ferrous sulphate. The ease of oxidation of iron(II) to iron(III) makes such baths unstable, so they would have to be either freshly prepared for each experiment or constantly monitored. Replacing Fe(II) with Fe(III) solves this problem, so the Fe(III) ion concentration is known more precisely. The choice of this particular ferric compound was justified by its high solubility and by Fe(III) ion already being in an appropriate citrate complex.

As noted before, the concentrations of particular inducing metal salts varied throughout the experiment. The utilized baths can be divided into four series, A, B, C and D, respecting their composition. The number of repetitions is mentioned in the brackets next to the series names. In series A (12 samples, 4 samples of each alloy) were baths containing one inducing metal salt, Fe(III), Co(II) or Ni(II), in a concentration of 27 mM, which produced the binary alloys WFe, WCo or WNi, respectively. Series B (12 samples, 4 samples of each alloy) consisted of baths with two (13.5 mM) or three (9 mM) inducing metal ions at equal concentrations totaling 27 mM altogether, leading to the deposition of the ternary alloys WFeCo, WfeNi and WcoNi, and the quaternary alloy WfeCoNi. Whereas the results of the deposition from bath series A and B allow a comparison of the deposition rates of particular iron-group metals in the same conditions, the remaining two series were set up to examine the influence of Cu on deposition rates in a high-entropy alloy. Series C (WfeCoNi, 11 samples) and D (WfeCoNiCu, 9 samples) consisted of baths designed to generate, respectively, quaternary and quinternary alloys possibly close to equal *at%* of the metals, i.e., equal deposition rates. The exact ion concentrations used in the C and D bath series are shown in Table 3. The experiments concerning the quaternary and the quinternary alloys were repeated a relatively large number of times, taking the system complexity into account.

During the electrodeposition, the plating bath was separated from 0.25 M sodium sulphate solution as an anode electrolyte. Two inert Ti/RuO_x_ anodes were placed symmetrically around the cathode. All the samples were deposited with a constant current at a current density *j* = 50 mA·cm^−2^. A potentiostat/galvanostat, EG&G PAR 173A, was utilized as the source of the current. All the experiments were conducted at a constant temperature of 65 °C. The coatings were plated on 0.999 Ag foil, cut into rectangles so that the area of the substrate was *A* = 2.4 cm^2^. The duration of the deposition was *t* = 1800 s. Before deposition, all the substrates were cleaned by annealing in a CH_4_–O_2_ flame, then mechanically scrubbed with a detergent and finally washed with ethanol. Each time, the clean Ag substrate was weighed on an analytical balance with a precision ±0.1 mg. After the deposition, each sample was weighed again in order to calculate the mass of the deposit Δ*m*.

The composition of the deposited alloys was measured using an FE SEM Zeiss Merlin (Jena, Germany) with a Bruker Quantax 400 EDS analyzer (Billerica, MA, USA) with a 10 mm^2^ detector. The EDS spectra were acquired at 15 kV beam energy and 400 pA beam current as an average from a (0.3 × 0.4) mm site in the very center of the sample. An example of the EDS spectrum for a WFeCoNiCu alloy sample is depicted in Figure 1a, with the characteristic lines labeled with element symbols. The numerical results were calculated from the acquired spectra using the original Bruker software (Quantax Esprit v.1.9.4) and normalized to 100% after omitting oxygen, the presence of which indicates surface oxidation of the alloy, and finally carbon, which might contaminate the surface before measurement. The oxygen content did not exceed a few percentage points in any sample. The metal content was quantified from the spectra using the K-series lines of Fe, Co, Ni and Cu, and the L-series lines of W (circled in Figure 1a). The mass content *wt%X* and atomic content *at% X* of the metals in the samples can be easily converted into each other, utilizing the molar masses M of all the metals present in the alloy. The morphology of the alloy surface was imaged with a secondary electron detector with the same beam parameters as the EDS measurements.

The XRD pattern was acquired with an X-ray diffractometer, Bruker D8 Discover, equipped with a collimated Cu Kα (1.54 Å) radiation source. The data were collected in the 20–110° 2θ range, with a 0.01° step-size, in a locked–coupled mode using a Vantec 1D-linear detector. The alloy sample prepared for the XRD analysis was deposited on a 15 × 15 mm piece of Ag foil as a 8 μm thick coating. The crystallite size was calculated using the Scherrer equation.

For each metal *X* (Fe, Co, Ni or Cu), the deposition rate [mol · cm^−2^ · s^−1^] calculated for a particular sample *v* is *X* = Δ*m* · *wt%X* · *A*^−1^ · M^−1^ · *t*^−1^, assuming that the electrodeposition process is not affected by changing *A* or *t*. The final results, i.e., the values of the *v* of the inducing metals, *at%* of tungsten and other metals, and *CE* of the whole process, were calculated as an average within the four mentioned bath series. All the following values of *v* have been multiplied by 10^5^ for legibility of the data.

## 3. Results

All the obtained alloys formed shiny gray metallic coatings that adhered well to the substrates. Only the WFe coating was observed to turn brownish after a longer exposition to air, which indicates that this alloy is more prone than the others to corrosive oxidation. The samples of the quinternary WFeCoNiCu alloy appeared to be slightly brighter in comparison to the others, which may be caused by the well-levelled surface morphology of the coating.

Figure 1b depicts an image of the WFeCoNiCu alloy surface, acquired with SEM. The surface consists of compact metallic structures, arranged in lines, along with some spherical granules grown on the surface. The surface is slightly cracked in quite a regular manner. The morphology of the WFeCoNiCu sample is typical of electrodeposited tungsten alloys and resembles other alloys such as WNiCu [25]. The presence of the cracks on the alloy surface is most probably caused by high tensile stress, as well as by abundant hydrogen evolution on the cathode surface during the electrodeposition, which is also the cause for the appearance of the linear grooves, parallel to the vertical direction during the electrolysis.

A diffractogram acquired for the WFeCoNiCu alloy sample is shown in Figure 1c. It consists mainly of an intense, broad peak at 2θ = 43.3°, and an additional reflection, weaker and even broader, at 2θ ca. 76°. The first one is typically the only peak observed for the electrodeposited alloys of tungsten with iron-group metals containing more than 18 at% W [2,15,19,27]. The full width at half maximum of this peak is 5.5°, which translates to a crystallite size of 1.62 nm, which means that the alloy is nanocrystalline, close to amorphous. A second, weaker reflection was also observed for ternary electrodeposited tungsten alloys containing copper [25]; thus it can be stated that the novel WFeCoNiCu alloy has an analogous structure to the previously researched tungsten–iron-group-metal–copper alloy coatings.

In Table 4, the numerical results for the binary alloys are shown, i.e., experiment series A. The presented data include the deposition rates of the inducing metals, atomic content of tungsten and faradaic efficiency of the deposition processes. All the rates here and subsequently are expressed in SI units [mol · m^−2^ · s^−1^] and multiplied by 10^5^ for legibility.

The efficiency of the deposition depends strongly on which metal was codeposited with the tungsten, and so do the deposition rates for particular metals. More precisely, the deposition rate of cobalt for the WCo alloy is about 4 times higher than that of nickel in the WNi alloy, and the deposition rate of iron in the WFe is about 10 times higher than that of nickel. In general, the tungsten content in all of these binary alloys is ca. 24%, but it tends to be slightly higher when the deposition is slower. Thus, out of these three alloys, the WNi has the highest W content and the WFe has the lowest in the present study.

In experiment series B, ternary and quaternary alloys were deposited, utilizing baths containing equal concentrations of the inducing metal salts. Therefore, compared to series A, the concentration of a particular metal was two times lower for a ternary alloy and three times lower for the quaternary WFeCoNi alloy. Table 5 contains the relevant data for the alloys deposited in series B.

The results from Table 4 and Table 5 are presented below as bar charts in Figure 2 and Figure 3. Figure 2 is a straight representation of the values of *v* for the Fe, Co and Ni deposition in particular binary, ternary and quaternary alloys. The values of the deposition rates, as given in Figure 2, are easily comparable visually within the categories of binary, ternary or quaternary alloys. However, as already mentioned, the particular inducing metal ion concentrations for the ternary metal deposition were divided by half, so the constant 27 mM concentration is shared by the two inducing metal ions in the bath, and so on with the bath for the quaternary alloy and three inducing metal ions. Thus, Figure 3 is presented as well, where the rates have been normalized to 1 mM metal ion concentrations, that is, divided by 27 for the binary alloys, by 13.5 for the ternary alloys and by 9 for the quaternary WFeCoNi alloy deposition. Presented in such a way, the differences between the values of the deposition rates can be compared for the particular inducing metals, especially in terms of divergencies from the linearity of *v* as a function of the metal ion concentration.

In general, during the deposition of the ternary alloys, iron still deposits at the highest rate, whilst nickel deposits the most slowly. In comparison to the WFe, the deposition rate of Fe is about 2 times lower for the WFeCo and WFeNi alloys, and approximately 3 times lower for the WFeCoNi, so it seems to be close to proportional to the iron concentration in the plating bath. However, the deposition rates for the remaining metals change in a different manner. As for Co, the deposition rates in the ternary alloys are noticeably higher than half the rate for the WCo; also, the deposition rate in the WFeCoNi is higher than a third of that for the binary alloy. The opposite can be observed for Ni: the rates of Ni deposition in the ternary and quaternary alloys are much lower than would be expected if they were proportional to the nickel concentration in the baths. For that reason, in the quaternary WFeCoNi alloy, the deposition rates of Fe and Co approach each other, so they are both more than 10 times higher than the rate of Ni deposition. The tungsten content in the ternary alloys containing nickel is close to that in the binary WNi, and for the WFeCo, it is very close to tungsten content in the WFe. In the WFeCoNi alloy, the tungsten content is somewhat in the middle, closest to *at%* W in the WCo alloy. In most cases, the faradaic efficiencies for the ternary and quaternary alloys are close to the mean values of the efficiencies for the binary alloys with corresponding metals.

Table 6 contains analogous data for experiment series C and D, i.e., the quaternary WFeCoNi and quinternary WFeCoNiCu alloys. As mentioned earlier, these baths contained adjusted concentrations of the metal ions, so to obtain alloys consisting of possibly equal *at%* of the inducing metals, also included in Table 6. Due to the *v* of nickel being more than 10 times lower than that of the other metals, the Ni(II) concentration had to be an order of magnitude higher than that of the other metals (see Table 3). Also, the proper concentration of Cu(II) was determined in a sequence of subsidiary tests between series C and D.

The values of the deposition rates contained in Table 6 are also depicted in Figure 4 in a visually comprehensive manner.

Apparently, the deposition rates of each inducing metal in both alloys are indeed close to equal, within standard deviation intervals. When comparing the WFeCoNi series B and C, it should be noticed that the tungsten content in the latter exceeds 30%, which is much higher than in any previous alloy in series A or B. On the other hand, the faradaic efficiency of the deposition process reaches only 5%, which is lower than for any other alloy except the binary WNi.

For the quinternary WFeCoNiCu alloy (series D), the tungsten content goes back to 24%, but all the rates of metal deposition are about 1.25 times higher than those without copper, including the *v* of tungsten. The deposition rate of Fe seems to be the most affected, and Co the least, although it is still clearly higher. Obviously, the copper itself deposits simultaneously with the other metals. Altogether, the introduction of copper makes the faradaic efficiency 1.5 times higher. Taking the standard deviation ranges into account, the obtained high-entropy WFeCoNiCu alloys have well-balanced atomic composition with 24% tungsten and ca. 19% of each inducing metal.

## 4. Discussion

A quinternary WFeCoNiCu alloy has been successfully synthesized utilizing the electrodeposition process. The morphology and crystalline structure of the alloy were shown to be comparable with previously researched binary and ternary tungsten alloys, especially the ternary alloys of tungsten, an iron-group metal and copper [25]. The novel quinternary alloy appears to be a homogeneous nanocrystalline material, close to amorphous. The XRD pattern (Figure 1c) proves that the WFeCoNiCu alloy contains no inclusions of other systems, such as metal oxides or solid solution phases of non-tungsten alloys. The issue of homogeneity of the electrodeposited tungsten alloys was already covered more exhaustively in our previous studies [27,28].

Out of the three most researched binary alloys of tungsten, i.e., with Fe, Co or Ni, the WFe is deposited with the highest faradaic efficiency and the WNi with the lowest, which was also observed in the cited papers. Overall, the achieved deposition rates are somewhat lower than the rates found in the literature (see Table 1), which may be caused by various factors such as discrepancies in the current densities, bath composition and temperature. However, for precise control of the composition of a quinternary alloy, knowledge of the ratios between the rates of electrodeposition in the same conditions is way more important than maximizing the rates themselves.

The current efficiencies are low, especially for nickel-containing alloys, compared to the much higher efficiencies of alloys such as WFe or WFeCo, which are an order of magnitude higher. Nevertheless, the low efficiency of the WNi alloy deposition does not stop it from being widely researched, due to its practical applications [6,10]. The particularly low metal deposition leads to abundant hydrogen evolution, which could cause issues with bath alcalization. For the current study, this issue was solved by utilizing fresh batches of the plating bath, but in other settings, the bath pH can be easily monitored and lowered in the case of the emergence of such a need.

In general, higher deposition rates of the inducing metals coincide with higher deposition rates of tungsten and a higher overall efficiency of the deposition process. According to the most recent mechanism proposal for tungsten codeposition [29], the [(*Me*)(WO_4_)(H)(Cit)]^n−^ type complex is a direct precursor of the electrodeposited tungsten alloy for *Me* being Ni or, most probably, other iron-group metals, namely, Co and Fe. Thus, the efficiency of the deposition process should be proportional to the concentration of the particular complex in the plating bath solution. Taking this into account indicates that the stability of these complexes present in the tungstate–citrate baths is increasing in the sequence *Me*: Ni < Co < Fe.

The copresence of two or three inducing metal ions in one plating bath at equal concentrations causes disproportionate changes in the deposition rates of the inducing metals. Most notably, the Ni deposition rate goes down in the presence of other inducing metals, which also could be explained by competition between the metal ions to form the adequate precursor complex.

Tuning the bath composition in a such manner that the metal deposition rates are equal inevitably requires that the metal ions in the solution be mostly nickel (about 88% of inducing metal ions in the bath). Hence, the faradaic efficiency is relatively low. However, even though the metal ions are mostly nickel, the efficiency of the deposition from such a bath is 2.5 times higher compared to the deposition of the binary WNi alloy. Also, in the quaternary alloy deposited from such a bath, the *at%* W is undoubtedly highest, exceeding 30%, which shows that this balanced setting is optimal in terms of the tungsten content. The addition of copper ions into the plating bath not only introduces copper into the alloy, but also catalyzes the deposition of every other metal, which leads to a remarkable boost in current efficiency. Questions about the exact role of copper in the mechanism of induced codeposition are yet to be answered in future studies.

Taking all the above into consideration should enable relatively precise control of Fe, Co, Ni and Cu content in the electrodeposited tungsten alloys of higher complexity, especially in the high-entropy quinternary WFeCoNiCu alloy. The deposition rates of the studied metals depend on their ion concentrations in a manner linear enough to modify the metal content predictably. Further iterations of adjusting the inducing metal ion concentrations may easily allow tuning of the composition of the high-entropy alloy, either for obtaining an excess of a chosen metal or to have the material composition well balanced. Due to the completely different behavior of tungsten species within the supposed codeposition mechanism, controlling the W content in the alloy would be a completely different task; thus was not included in the present study.

## Figures and Tables

**Figure 1 materials-17-01513-f001:**
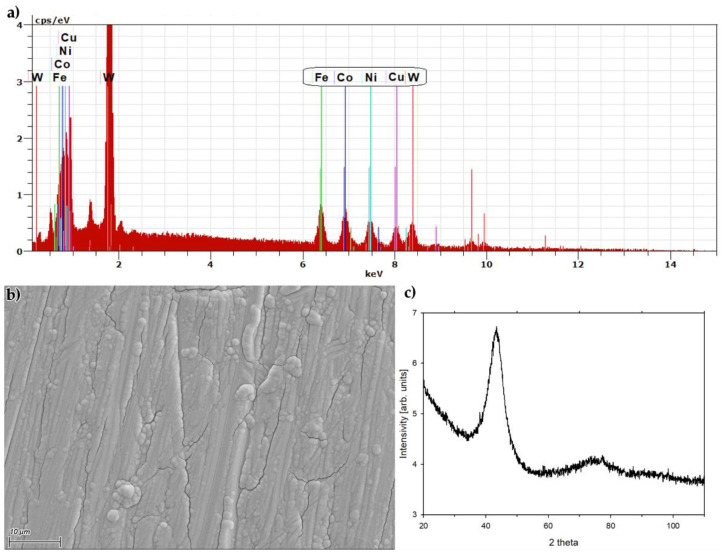
(**a**) EDS spectrum acquired for a WFeCoNiCu alloy sample. (**b**) SEM image of WFeCoNiCu coating surface. Scalebar in the bottom left corner. (**c**) XRD pattern for WFeCoNiCu alloy sample.

**Figure 2 materials-17-01513-f002:**
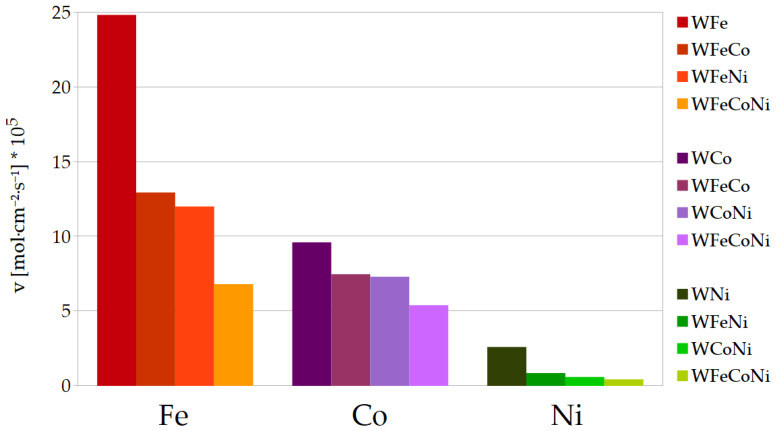
Comparison of Fe, Co and Ni deposition rates in binary, ternary and quaternary alloys.

**Figure 3 materials-17-01513-f003:**
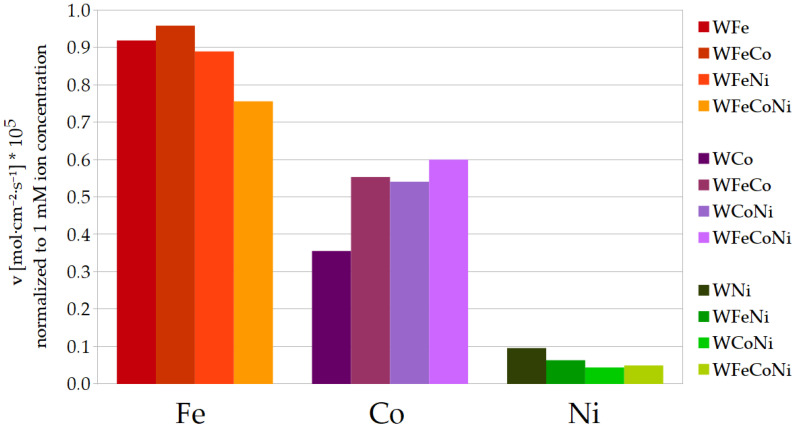
Metal deposition rates normalized to 1 mM metal ion concentration.

**Figure 4 materials-17-01513-f004:**
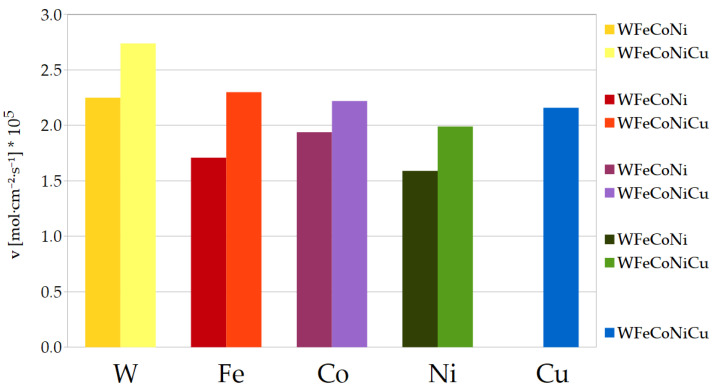
Influence of Cu on deposition rates in multinary alloys.

**Table 1 materials-17-01513-t001:** Rates of Fe, Co and Ni codeposition with tungsten, according to cited papers. Numbers from sources marked with “*” have been retrieved from graphs.

Alloy	WFe			WCo			WNi		
Source	[11] *	[12]	[13]	[14]	[15] *	[16] *	[17]	[18]	[19] *
*j* [mA·cm^−2^]	100	100	100	35	36	50	15	100	15
*CE* [%]	27.8	15	39.7	15.3	74	53	8	3.75	11.1
*at%* W	6.3	29.4	38.6	38.2	22.3	30.5	31	34.7	46.4
*at%* Fe/Co/Ni	93.7	70.6	61.4	61.8	77.7	69.5	69	65.3	53.6
*v* Fe/Co/Ni	135	36.6	101	17.1	107	95.4	4.29	12.7	4.63

**Table 2 materials-17-01513-t002:** Composition of the plating bath.

Reagent	Concentration
Sodium citrate	270 mM
Sodium tungstate	260 mM
Boric acid	170 mM
Phosphoric acid	90 mM
Nonoxynol-10 (surfactant additive)	70 ppm
Butynediol (brightening additive)	50 ppm
Distilled water (Millipore Milli-Q)	—
Nickel sulphate Cobalt sulphate Ferric ammonium citrate Copper sulphate	various concentrations, totaling 27 mM, explained further in detail

**Table 3 materials-17-01513-t003:** Concentrations of metal ions in C and D bath series.

Bath Series	C Fe(III)	C Co(II)	C Ni(II)	C Cu(II)
C	1.55	1.70	23.75	—
D	1.55	1.70	23.75	1.55

**Table 4 materials-17-01513-t004:** Comparison of deposition rates for binary alloys (series A).

Alloy	WFe	WCo	WNi
Me	Fe	Co	Ni
*v* Me	24.8 ± 3.0	9.6 ± 2.5	2.60 ± 0.69
*v* W	7.20 ± 0.98	3.1 ± 1.1	0.87 ± 0.21
*at%* W	22.5 ± 1.3	24.0 ± 2.0	25.3 ± 2.0
*CE* [%]	24.9 ± 3.9	7.3 ± 2.2	2.01 ± 0.50

**Table 5 materials-17-01513-t005:** Comparison of deposition rates for ternary and quaternary alloys (series B).

Alloy	*v* Fe	*v* Co	*v* Ni	*v* W	*at%* W	CE [%]
WFeCo	12.93 ± 0.37	7.47 ± 0.89	—	5.73 ± 0.28	21.96 ± 0.83	17.01 ± 0.72
WFeNi	12.0 ± 3.3	—	0.86 ± 0.11	4.44 ± 0.69	26.0 ± 2.0	12.4 ± 2.7
WCoNi	—	7.3 ± 1.0	0.60 ± 0.11	2.75 ± 0.50	25.6 ± 1.1	6.2 ± 1.0
WFeCoNi	6.8 ± 1.9	5.4 ± 1.0	0.447 ± 0.057	4.07 ± 0.76	24.43 ± 0.88	10.9 ± 2.3

**Table 6 materials-17-01513-t006:** Comparison of deposition rates for quaternary and quinternary alloys (series C and D).

**Alloy**	***v* Fe**	***v* Co**	***v* Ni**	***v* Cu**	***v* W**	**CE [%]**
WFeCoNi	1.71 ± 0.22	1.94 ± 0.17	1.59 ± 0.23	—	2.25 ± 0.15	4.95 ± 0.33
WFeCoNiCu	2.30 ± 0.42	2.22 ± 0.37	1.99 ± 0.37	2.16 ± 0.34	2.74 ± 0.38	7.7 ± 2.8
**Alloy**	***at%* Fe**	***at%* Co**	***at%* Ni**	***at%* Cu**	***at%* W**	
WFeCoNi	22.8 ± 2.0	25.9 ± 1.9	21.3 ± 3.3	—	30.05 ± 0.42	
WFeCoNiCu	20.1 ± 1.9	19.4 ± 1.4	17.3 ± 1.7	19.3 ± 1.8	23.96 ± 0.76	

## Data Availability

The original contributions presented in the study are included in the article; further inquiries can be directed to the corresponding author.

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
