# Peer review of "Comparative Analysis of Metal Electrodeposition Rates towards Formation of High-Entropy WFeCoNiCu Alloy"

_materials, 2024, doi:10.3390/ma17071513_

Round 1

Reviewer 1 Report (Previous Reviewer 1)

Comments and Suggestions for Authors

In my opinion, the authors have clarified all the questions posed in previous review and the manuscript could be published.

Author Response

Dear Reviewer,

thank you kindly for the review and the very constructive questions regarding our work, which have lead to some major improvements.

Best regards,

Authors

Reviewer 2 Report (Previous Reviewer 3)

Comments and Suggestions for Authors

The current version of this paper is not recommended for publication. It is unclear how relevant the issue of comparing the electroplating speeds of metals from different sources is to the scientific community and practical applications. Previous studies have already conducted experiments on the electrodeposition of alloys. There is no compelling evidence that the results of this work significantly contribute to the existing knowledge in this field. Specific practical applications of the obtained results and their utilization methods are not presented. Furthermore, the text contains an excessive amount of information in the Introduction, incorrect formulation of the research aim and objectives, lack of information on experiment repeatability, insufficient details about the equipment used and analysis methods employed, and inconsistency between the conclusions and the obtained results. Additionally, a significant concern arises from the lack of coherent data presented in the study, particularly the absence of meaningful SEM and XRD results.

Author Response

Please find the statements of yours enlisted and accompanied by meticulously argumented answers in the file below. We are not able to apply any further changes on basis of mere opinions.

Reviewer 3 Report (New Reviewer)

Comments and Suggestions for Authors

The authors described the electrodeposition rate of different tungsten alloy materials. This study is interesting as it could bring new insight into the electrodeposition of tungsten alloy materials. Some concerns need to be addressed before accepting the manuscript.

1. The deposition rate of co-electrodeposition alloy could be dependent on temperature. The authors did not explain why high temperature is used for electrodeposition.

2. The materials and methods part should be more specific to reproduce the data or follow the work. How about stirring the solution?  The solution convection effect may influence the homogeneous metal film formation.

3. What is the metal film thickness for each alloy system?

4. How about the elemental mapping of EDS analysis for quaternary WFeCoNiCu alloy composition? What is the Oxygen percentage?

5. Why the Faradic efficiency (CE) is low? What are the possible reasons for such low CE ?

6.   Is the quaternary WFeCoNiCu alloy composition similar throughout the film thickness? How about an in-depth XPS or GDOES study? What is the oxygen content for each alloy system?

7. What is the mechanism for forming quaternary WFeCoNiCu alloy by electrodeposition?

Comments on the Quality of English Language

English need to be improved. Some sentences in the result section are too long to understand.

Author Response

Dear Reviewer,

please find a file below with very exhaustive answers to your questions. We have applied a few additions to our manuscript, according to your suggestions. We hope that the attached answers will clarify all your doubts, without putting all the information in the core manuscript. We are eager to answer any further questions concerning the subject of our work.

Best regards,

Authors

Round 2

Reviewer 3 Report (New Reviewer)

Comments and Suggestions for Authors

The authors descibed all the concerns. The manuscript can be accepted.

This manuscript is a resubmission of an earlier submission. The following is a list of the peer review reports and author responses from that submission.

Round 1

Reviewer 1 Report

Comments and Suggestions for Authors

The manuscript investigates the role of different inducing  metals (Fe, Co, Ni, Cu) for W electrodeposition, on the current efficiency of W-alloy electrodeposition from aqueous baths. The research is interesting because in paves the way for further formation of novel ternary and quaternary alloys. The experimental work and discussion are solid. Prior to publication, it would be useful if the author answered the following questions: 

The term "high entropy alloy" is used throughout the text. It would be useful to define this term in the text. Based on which properties the coating obtained is called high entropy alloy?

In line 248, authors state that "it should indicate that the complex stability constant is increasing in sequence Me = Ni < Co 248 < Fe." Please support and clarify this statement better. The opposite would be more logical on the first glance to the common reader who is not familiar with the mechanism: higher complex stability gives lower deposition rate, and vice versa. For this reason, short description of the electrodeposition mechanism would be more useful here, than just calling the reference. 

Low current efficiencies (sometimes lower than 3%) indicate enormous hydrogen evolution reaction. Some paragraphs should be dedicated to this issue. How can the authors be sure that some or all metals are not deposited in the form of hydroxides, instead of in the pure state? What is the pH increase, can it be calculated? Does it effect the electrodeposition mechanism?

Reviewer 2 Report

Comments and Suggestions for Authors

There are no micrographs and macrographs presented to enable one to further evaluate the depth of the work. Just a couple of calculation results about deposition rates, of course are not sufficient for a paper to be entitled for publication in an international journal. The work is shallow and superficial. The experimental procedures are quite vague and short with no details. The evidences provided for the claims propounded in Discussion are marginal and trivial. Despite English is generally good, in many instances it needs to be edited, reviewed and improved. This article with the present form is not suitable for publication in Materials and its rejection is recommended.

Comments on the Quality of English Language

Despite English is generally good, in many instances it needs to be edited, reviewed and improved.

Reviewer 3 Report

Comments and Suggestions for Authors

I have carefully reviewed the article "Comparative analysis of metal electrodepositon rates towards formation of high-entropy WFeCoNiCu alloy" submitted for review and regretfully must report that in its current form, this work cannot be recommended for publication.

Primarily, concerns exist regarding the timeliness of the study presented. While the authors claim a need to conduct original experiments to solve the problem of comparing deposition rates from disparate sources, the extent to which this truly presents an issue meriting attention from the scientific community and practical applications remains undefined.

Secondly, the novel contribution of this work is not readily apparent. Though experiments depositing alloys of WFe, WCo, WNi and others at varying concentrations were performed, comparable investigations have previously been undertaken. Convincing evidence is not furnished to demonstrate the results herein substantially augment existing knowledge in this domain.

Thirdly, doubts surround the practical significance of the findings. The authors assert their data will facilitate more precise composition control of multi-element W alloys but specific practical applications are not considered. Unclear is where and how the results may productively be employed.

Deficiencies in the work additionally include:

– an introduction containing surplus background immaterial to the study presented;

– objectives and aims formulated incorrectly;

– lack of information on experimental repetition;

– insufficient technical details on equipment and analytical methods;

– conclusions not fully aligned with outcomes obtained;

– stylistic issues with the text; outdated source material comprising the nineteen-item bibliography, none published within the past five years for such an advanced field being inappropriately deficient;

furthermore, the study includes no material structure characterization data (SEM, EDS, XRD);

Therefore, owing to concerns regarding relevance, lack of demonstrable novelty, ambiguous usefulness, and the numerous comments needing addressed, this article cannot be recommended for publication in its present form. The authors would be wise to thoroughly rework the manuscript considering the feedback provided and resubmit it for re-review.